# Mechanochemical Synthesis and Nitrogenation of the Nd_1.1_Fe_10_CoTi Alloy for Permanent Magnet

**DOI:** 10.3390/molecules26133854

**Published:** 2021-06-24

**Authors:** Hugo Martínez Sánchez, George Hadjipanayis, Germán Antonio Pérez Alcázar, Ligia Edith Zamora Alfonso, Juan Sebastián Trujillo Hernández

**Affiliations:** 1Departamento de Física, Universidad del Valle, Meléndez, Cali A.A. 25360, Colombia; german.perez@correounivalle.edu.co (G.A.P.A.); ligia.zamora@correounivalle.edu.co (L.E.Z.A.); juan.trujillo@unibague.edu.co (J.S.T.H.); 2Department of Physics and Astronomy, University of Delaware, 217 Sharp Lab, Newark, DE 19716, USA; hadji@udel.edu; 3Centro de Excelencia en Nuevos Materiales (CENM), Universidad del Valle, Meléndez, Cali A.A. 25360, Colombia; 4Facultad de Ciencias Naturales y Matemáticas, Universidad de Ibagué, Ibagué 730007, Colombia

**Keywords:** permanent magnets, mechanochemical synthesis, nitrogenation, extrinsic magnetic properties, X-ray diffraction patterns, spin reorientation

## Abstract

In this work, the mechanochemical synthesis method was used for the first time to produce powders of the nanocrystalline Nd_1.1_Fe_10_CoTi compound from Nd_2_O_3_, Fe_2_O_3_, Co and TiO_2_. High-energy-milled powders were heat treated at 1000 °C for 10 min to obtain the ThMn_12_-type structure. Volume fraction of the 1:12 phase was found to be as high as 95.7% with 4.3% of a bcc phase also present. The nitrogenation process of the sample was carried out at 350 °C during 3, 6, 9 and 12 h using a static pressure of 80 kPa of N_2_. The magnetic properties M_r_, µ_0_H_c_, and (BH)_max_ were enhanced after nitrogenation, despite finding some residual nitrogen-free 1:12 phase. The magnetic values of a nitrogenated sample after 3 h were M_r_ = 75 Am^2^ kg^–1^, µ_0_H_c_ = 0.500 T and (BH)_max_ = 58 kJ·m^–3^. Samples were aligned under an applied field of 2 T after washing and were measured in a direction parallel to the applied field. The best value of (BH)_max_ ~ 114 kJ·m^–3^ was obtained for 3 h and the highest µ_0_H_c_ = 0.518 T for 6 h nitrogenation. SEM characterization revealed that the particles have a mean particle size around 360 nm and a rounded shape.

## 1. Introduction

The high demand for Nd_2_Fe_14_B permanent magnets, with maximum energy product (BH)_max_ up to 450 kJ·m^–3^ [1], in applications such as electric vehicles and wind turbines, implies a high production cost owing to the considerable content of Nd. The ferromagnetic R(Fe,M)_12_ compounds, where R is a rare earth element and M is a stabilizing element (V, Ti, Mo, Cr, W, Al or Si), are currently being investigated as alternatives to Nd-Fe-B due to their excellent magnetic properties and their much lower R content [2,3]. It has been found that the ferromagnetic R(Fe,M)_12_ compounds crystalize in the tetragonal ThMn_12_-type structure (space group I4/mmm) with R= Y, Nd, Sm, Gd, Tb, Dy, Ho, Er, Tm or Lu [4], and that these compounds exhibit strong exchange interactions for the light rare earths. The Sm(Fe,M)_12_ compounds are characterized by a strong uniaxial anisotropy [3] and develop a coercive force µ_0_H_c_ as high as 1.17 T [5]. The second type of attractive compounds are the Nd(Fe,M)_12_Nx and Pr(Fe,M)_12_Nx with M = Ti, V or Mo [6]. The highest coercivity values for these compounds with a subsequent nitrogenation, µ_0_H_c_ up to 1.3 T, have been achieved in nanocrystalline quasi-isotropic alloys with M = Mo [7,8,9,10]; somewhat lower µ_0_H_c_ values, up to 0.9 T, were reported for nanocrystalline alloys with M = V [11]. Although it is most favorable for a high coercivity, the microstructure of randomly oriented nanograins—usually generated through melt-spinning or high-energy ball-milling—dramatically lowers the attainable (BH)_max_ due to isotropic nature of the samples. The same alloys (M = Mo, V) also allowed for the synthesis of anisotropic R(Fe,M)_12_N_x_ powders with µ_0_H_c_ of 0.35–0.58 T [12,13,14,15,16]. No significant coercivity, however, could be developed in the NdFe_11_TiN_x_ compound, which is particularly attractive due to the high M_s_ of 1.48 T [17]; even the µ_0_H_c_ reported for the nanocrystalline alloys did not exceed 0.23 T [11]. The mechanochemical synthesis was demonstrated to yield single-crystal submicron and nanoparticles of hard magnetic materials, including the R(Fe,M)_12_ alloys, which exhibit coercivity values typical of nanocrystalline powders [18,19]. This synthesis technique also requires less-expensive oxides, rather than metals, as the raw materials, and it may be expected to reduce local demagnetization fields due to more regular shapes of the particles [20]. The authors are not aware of any reports on mechanochemically synthesized R(Fe,M)_12_N_x_ compounds. By reducing nanoscale oxide precursors Su et al. [21] prepared anisotropic NdFe_10_Mo_2_N_x_ particles, which were a few microns in size and exhibited an µ_0_H_c_ of 0.35 T. The aim of this work was to prepare a Nd(Fe,Co,Ti)_12_N_x_ permanent magnet material through mechanochemical synthesis and subsequent nitrogenation of the Nd(Fe,Co,Ti)_12_ compound and to study its extrinsic magnetic properties as a function of the temperature and magnetic alignment.

## 2. Materials and Methods

### 2.1. Sample Preparation

The alloy with nominal composition Nd_1.1_Fe_10_CoTi was obtained by mechanochemical synthesis, using powders of Nd_2_O_3_ (nominal purity of 99.9%), Fe_2_O_3_ (particles ≤5 µm and 99% purity), TiO_2_ (≤5 µm and 99% purity) and Co (1.6 µm and 99.8% purity) as the raw materials. The reactants were mixed with 1 mm granules of Ca (99.5% purity) serving as a reducing agent and CaO powder acting as a dispersant. The amounts of the precursor powders were optimized using a mole ratio (Fe+Co+Ti)/Nd of 6.62 (necessitated by incomplete reduction of the rare earth oxide), and a Ca/O ratio of 2.0 (not counting the CaO dispersant). The mass of the CaO dispersant was three times the mass of the reactants other than Ca. The reactants were first mixed thoroughly with Ca and then with the dispersant. After that, a 5 g charge of the mixed powders was mechanically activated with a Spex-8000 high-energy (University of Delaware, Department of Physics and Astronomy, Sharp Lab, Newark DE, USA) mill by milling for 4 h in argon atmosphere, with six 12 mm steel balls. The subsequent annealing at temperatures from 950 to 1100 °C during 10 min was done in argon-filled quartz capsules, without exposing the as-milled powder to the air. A temperature of 1000 °C was found to be optimal for the formation of the ThMn_12_-type structure. For nitrogenation, the annealed powders were sealed in quartz capsules under 80 kPa of a 99.999%-pure N_2_ and held at 350 °C for 3, 6, 9 and 12 h. Finally, a multistep procedure of repeated washing with deionized water, glycerol, acetic acid, and ethanol [19] was used to collect the ferromagnetic nitrogenated particles, and to remove the CaO, rare earth oxides and unreacted Ca.

### 2.2. Powder Characterization

The powders were characterized by X-ray diffraction (XRD) in a Rigaku Ultima IV diffractometer (University of Delaware, Department of Physics and Astronomy, Sharp Lab, Newark, DE, USA) with Cu-Kα radiation, and the diffraction patterns were measured in the 2θ range from 25 to 60 degrees. The analysis of XRD patterns was done using the Maud program. Scanning electron microscopy (SEM) was used to study the particle shape and size distribution, using a JEOL JSM-6335F scanning electron microscope (University of Delaware, Department of Physics and Astronomy, Sharp Lab, Newark, DE, USA). The magnetic properties were measured using a Quantum Design VersaLab vibrating sample magnetometer (University of Delaware, Department of Physics and Astronomy, Sharp Lab, Newark DE, USA); the particles were immobilized with paraffin wax, in some cases while applying a 2 T orienting field (hereinafter, “aligned” powders). Correction for self-demagnetization was done using demagnetization factors experimentally determined for similarly prepared Ni powder.

## 3. Result and Discussion

### 3.1. Structural and Magnetic Characterization before Washing

After the mechanochemical activation and annealing at 1000 °C for 10 min, the powders were characterized by X-ray diffraction and vibrating sample magnetometry. The X-ray diffraction pattern of Figure 1 shows the peaks of the crystalline phases present before washing the powder. The highest peaks correspond to the CaO phase with cubic crystal structure (space group Fm-3m); the phase with ThMn_12_-type crystal structure (space group I4/mmm) or “1:12”, is present as a minority phase. The cubic bcc phase (space group Im-3m); and some peaks of the hexagonal Nd_2_O_3_ phase (space group P–63/mmc), as well as of the orthorhombic CaTiO_3_ phase (space group Pnma) were also identified. The peak at 2θ = 27.59 deg possibly corresponds to a rare earth oxide isotypical to Pr_24_O_44_ (space group P–1).

The hysteresis loop of the annealed and not washed powder is shown in Figure 2. Magnetization values at 3 T of 15.67 Am^2^kg^−1^, remanence M_r_ = 4.43 Am^2^kg^−1^, and µ_0_H_c_ = 0.121 T were obtained by analysis of this measurement. The magnetization values are low due to the predominance of the diamagnetic CaO diluting the ferromagnetic phase-(s). The low M_r_ and µ_0_H_c_ values are consistent with the weak anisotropy of nitrogen-free Nd(Fe,M)_12_.

### 3.2. SEM Characterization

Figure 3 shows the SEM images of the washed particles nitrogenated for different times.

The ferromagnetic particles tend to have rounded shape and a log-normal size distribution with a positive asymmetry (Fisher asymmetric coefficient γ_1_ > 0). The log-normal fit showed mean particle sizes of 0.29, 0.37, 0.35, 0.39, and 0.41 µm for the 0, 3, 6, 9, and 12 h nitrogenated samples, respectively. These results show that the particles tend to grow with the increase of nitrogenation time. It is worth noting that the small particle size of about 360 nm and the rounded shape of the particles are important factors for the decrease of the demagnetized field [20] and the improvement of the coercive force.

### 3.3. XRD after Washing

In Figure 4, the XRD diffraction patterns of the washed powders are shown. For the sample without nitrogenation (0 h), the pattern of the peaks of the 1:12 phase, and a single peak of the bcc phase at 2θ = 44.62 deg are present. The refinement of the pattern gave a volume fractions of 95.7 and 4.3% for the 1:12 and the bcc phases, respectively. The refined lattice parameters of the 1:12 phase were a = 8.589 Å and c = 4.808 Å. The average crystallite sizes found from the XRD refinement were 269.7(15.2) nm and 46.4(6.2) nm for the 1:12 and bcc phases, respectively. The mean crystallite size of the 1:12 phase is very close to the value of 0.29 µm obtained for the mean particle size (SEM), indicating that the particles of this phase are single crystals. These single crystal particles lead an improved μ_0_H_c_ because of their submicron size.

The XRD diffractions patterns for the samples nitrogenated for 3, 6, 9 and 12 h do not feature the characteristic shift of the peaks to the left. Instead, the peaks are broadened to their left side as it can be seen in Figure 4. This, apparently, indicates that in all the samples a certain part of the 1:12 phase remains nitrogen-free, even though most of the phase does absorb nitrogen. Thus, the resulting powders may be characterized by a gradient in the N content. In order to verify this hypothesis, Le Bail [22] analysis was carried out, assuming one bcc phase and three distinct 1:12 phases 1:12(a), 1:12(b) and 1:12(c). This analysis allowed for a good refinement of the XRD patterns with the parameters which are presented in Table 1.

The 1:12(a) phase is inherited from the parent, nitrogen-free alloy, because its lattice parameters and unit cell volume are similar, see Table 1. Its XRD peaks are in the right side of the XRD spectrum, as they do not shift to the left upon the nitrogenation. The 1:12(b) and 1:12(c) phases have bigger lattice parameters and therefore bigger volumes of nitrides, indicating that they absorbed different amounts of nitrogen. At the same time, the intensity of the bcc phase peak increases with the nitrogenation time (see Figure 4), indicating that the volume fraction of this phase increases from 4.3 to 8.5% when the nitrogenation time increases from 0 to 12 h.

### 3.4. Magnetic Characterization of Aligned and Non-Aligned Powders

Figure 5 presents the hysteresis loops of the randomly oriented Nd_1.1_Fe_10_CoTiN_x_ powders for 0, 3, 6, 9, and 12 h of nitrogenation. The M_3T_, M_r_ and µ_0_H_c_ magnetic properties for the 0 h sample increase after washing from 15.67 to 121.92 Am^2^kg^−1^, 4.43 to 57.79 Am^2^kg^−1^, and 0.121 to 0.276 T, respectively, due to the removal of non-magnetic CaO and the rare earth oxides. Moreover, the M_r_ and the µ_0_H_c_ values increase after nitrogenation, as it can be seen in the Figure 5 and are listed in Table 2 (in parenthesis). This enhancement of the magnetic properties is due to the interstitial modification of those 1:12 crystallites which absorb nitrogen, leading to enhancement of their magnetocrystalline anisotropy.

The values of µ_0_H_c_ decrease when the nitrogenation is increased from 3 to 12 h, apparently, because of a nearly doubled amount of the soft magnetic bcc phase. The M_r_ = 75.27 Am^2^kg^−1^, µ_0_H_c_ = 0.500 T, and (BH)_max_ = 58.38 kJ·m^−3^ values for 3 h of nitrogenation are comparable to those reported in [9,23], for isotropic nanocrystalline Nd(Fe,Mo)_12_N_x_ alloys, even though 1:12 alloys stabilized with Mo, are known to exhibit a higher coercivity than the 1:12 alloys stabilized with Ti.

In Figure 6, the hysteresis loops for the anisotropic ferromagnetic particles aligned with an external field are shown; the corresponding M_3T_, M_r_, µ_0_H_c_, and (BH)_max_ dates are listed in Table 2 (without parenthesis). All the hard magnetic properties were found to be enhanced by the alignment. It is obvious that the increase of M_3T_ and M_r_ is the immediate result of the easy axes of the crystallites being aligned with the direction of the measurements. The improvement of µ_0_H_c_ is caused by the increase of the magnetocrystalline anisotropy due to the nitrogenization. The reduced remanence M_r_/M_3T_ increases upon nitrogenation and the alignment; for 3, 6 and 9 h of nitrogenation, a value of 0.90 was obtained despite the soft magnetic bcc phase. The possibility to align the easy axes of magnetization in the Nd(Fe,M)_12_N_x_ particles is important for the development of high (BH)_max_ values, as it is shown in Table 2. A (BH)_max_ = 113.74 kJ·m^−3^ that was obtained for the aligned particles after nitrogenation for 3 h is the highest (BH)_max_ value reported so far for the Nd(Fe,M)_12_N_x_ compounds, and it contributes to closing the gap between the hexaferrites and the “RE-lean” magnets [1]. Moreover, this (BH)_max_ value is better than those reported for NdFe_9.4_Co_1.6_MoN_x_ [9], Mn-Al-C [24], and MnBi [25], that are 56.50, 73.21, and 70.82 kJ·m^−3^, respectively.

### 3.5. Magnetic Characterization at Low Temperatures

To study the evolution of M_3T_, M_r_, and µ_0_H_c_ with the temperature, the hysteresis loops were measured in the first and the second quadrant, for the sample nitrogenated at 6 h, as can see in Figure 7. The 6 h nitrogenation was selected because it exhibited one of the best magnetic properties after nitrogenation without alignment. The strong uniaxial anisotropy of the R(Fe,Ti)_12_ compounds has been associated with the strong coupling of both rare earth and transition metal magnetic sublattices, which favor an axial orientation [26]. However, experimental and theoretical studies have shown that for the NdFe_11_Ti(N or H) compounds, a rotation of the easy axis anisotropy to a canted or basal orientation with the decrease of the temperature is obtained, and this behavior is known as spin reorientation [4,27,28].

Temperature dependence of magnetization of the NdFe_11_Ti compound reported a peak at ~200 K related to the spin reorientation of this compound [4]. In aligned powders, a maximum of magnetization was observed at temperatures higher than the spin reorientation temperature [29]. The decrease of M(H) as a function of temperature shown in Figure 7 can be associated with the spin reorientation. The jumps of magnetization emerging in the 1st M(H) quadrant with the decrease of temperature can also be attributed to a metamagnetic spin reorientation transition [30]. The M_3T_, M_r_, and µ_0_H_c_ values are shown in Figure 8 as a function of the temperature.

For the µ_0_H_c_, a tendency to increase with the decrease in the temperature is shown in Figure 8; A maximum value of µ_0_H_c_ = 1.259 T was obtained at 100 K. This fact indicates that the spin-reorientation transition occurs around this temperature, in agreement with the reported spin-reorientation temperature for the NdFe_11_TiH compound (100 K [27]). On the other hand, this change in the µ_0_H_c_ with the decrease of the temperature was also observed in the (Nd_0.2_Ce_0.8_)_2_Fe_14_B compound at 100 K, where it was similarly associated with the spin-reorientation temperature [31]. For the lower temperature of 50 K, the µ_0_H_c_ decreases to 1.171 T as can be seen in Figure 8.

## 4. Conclusions

The mechanochemical synthesis has been demonstrated to be a powerful technique to synthetize submicron single crystal Nd(Fe,Co,Ti)_12_N_x_ particles with rounded shape, and reasonably good magnetic properties. The nitrogenated Nd_1.1_Fe_10_CoTiN_x_ compounds exhibited a significant improvement of their magnetic properties after being washed, and aligned, even though according to the Le Bail analysis of the XRD patterns about 40 vol.% of the original 1:12 phase, did not adsorb nitrogen in this experiment. The increase in the magnetic properties M_r_, µ_0_H_c_ and (BH)_max_ upon nitrogenation results in values of 75 Am^2^kg^−1^, 0.500 T and 58 kJ·m^−3^, respectively, for the sample with 3 h of nitrogenation. Comparison of the XRD analysis of the crystallite size and SEM characterization of the particles revealed that the washed 1:12 particles were monocrystalline. The alignment of the samples with an external magnetic field appears to be a very important factor to significantly enhance the magnetic properties. The best (BH)_max_ ~ 114 kJ·m^−3^ was measured for the sample with 3 h of nitrogenation, thus significantly improving the values previously reported for other systems. The best µ_0_H_c_ = 0.518 T was obtained in a sample with 6 h of nitrogenation. Low temperature M(H) curves show magnetization jumps, which indicate a first order magnetization process.

## Figures and Tables

**Figure 1 molecules-26-03854-f001:**
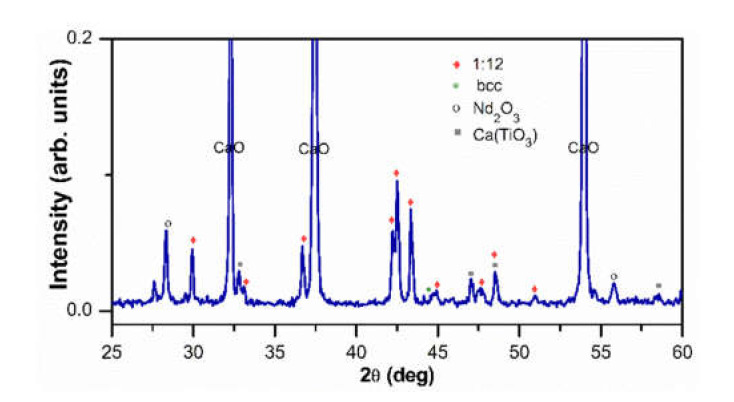
X-ray diffraction pattern of the Nd_1.1_Fe_10_CoTi powder annealed at 1000 °C for 10 min.

**Figure 2 molecules-26-03854-f002:**
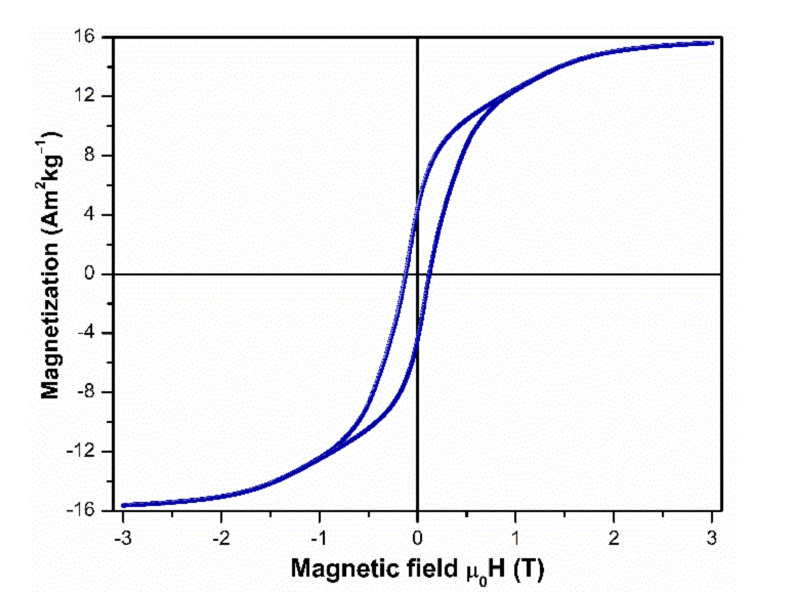
Hysteresis loop of the Nd_1.1_Fe_10_CoTi powder annealed at 1000 °C for 10 min.

**Figure 3 molecules-26-03854-f003:**
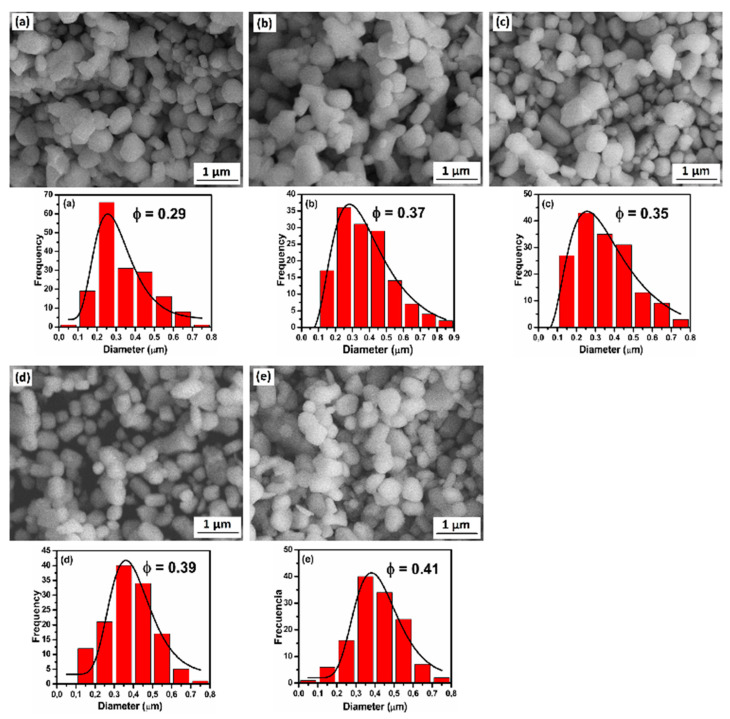
SEM micrographs of mechanochemical synthesized of Nd_1.1_Fe_10_CoTi particles nitrogenized for (**a**) 0 h, (**b**) 3 h, (**c**) 6 h, (**d**) 9 h and (**e**) 12 h.

**Figure 4 molecules-26-03854-f004:**
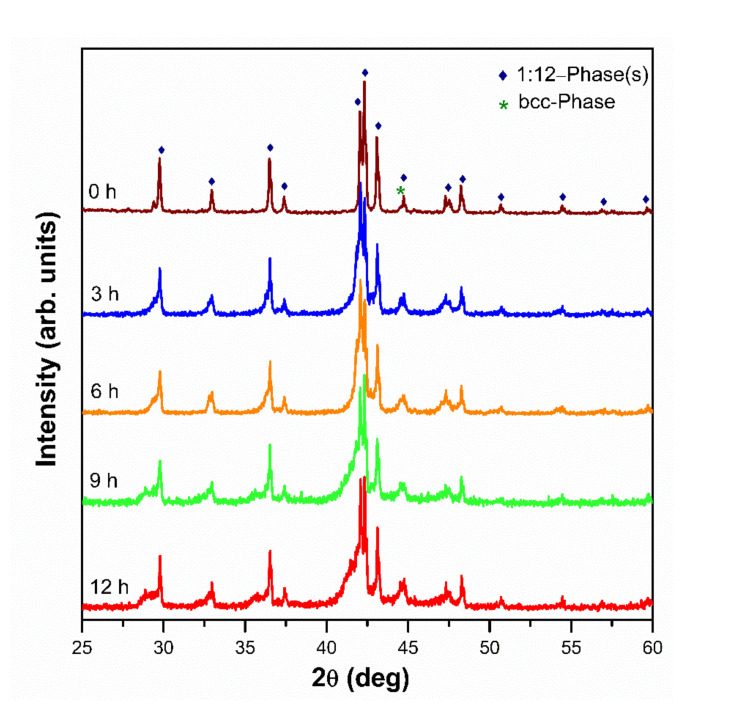
X-ray diffraction patterns of Nd_1.1_Fe_10_CoTi alloys nitrogenated from 0 to 12 h followed by washing.

**Figure 5 molecules-26-03854-f005:**
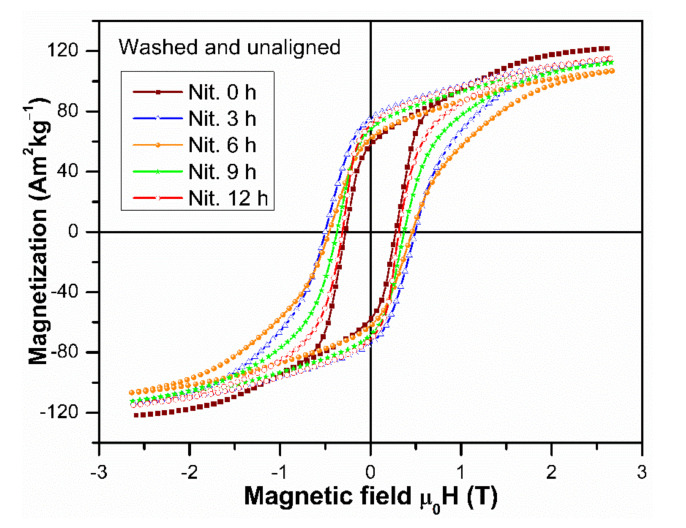
Hysteresis loops of the randomly oriented Nd_1.1_Fe_10_CoTiN_x_ alloys after washing as a function of nitrogenation time.

**Figure 6 molecules-26-03854-f006:**
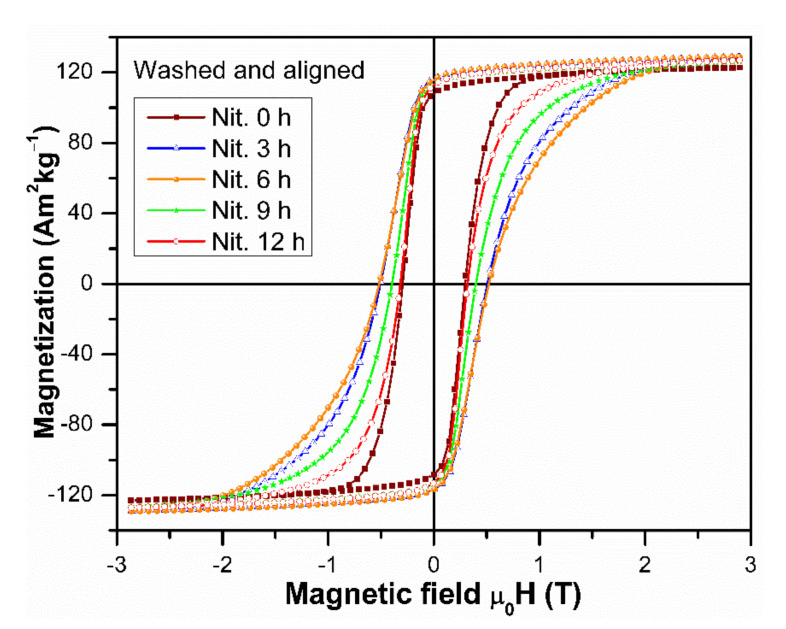
Hysteresis loops of the Nd_1.1_Fe_10_CoTiN_x_ alloys nitrogenized for the indicated time, washed, and magnetically oriented.

**Figure 7 molecules-26-03854-f007:**
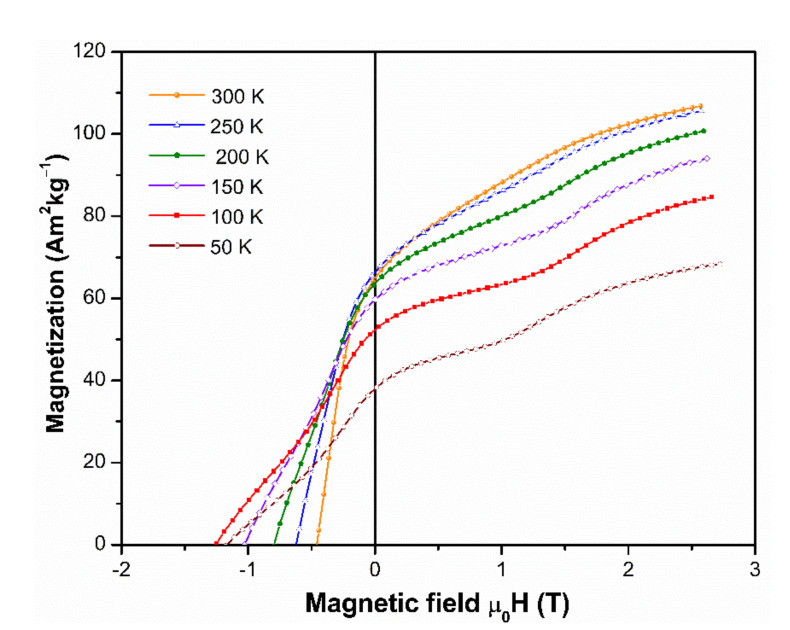
Demagnetization curves of Nd_1.1_Fe_10_CoTiN_x_ powder nitrogenated during 6 h and washed. The specimen was not magnetically oriented.

**Figure 8 molecules-26-03854-f008:**
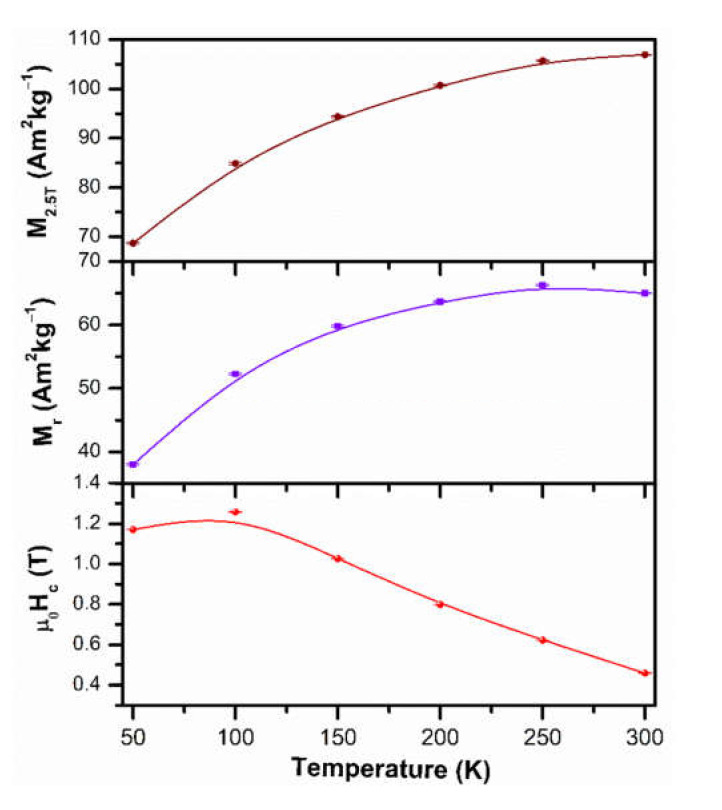
Saturation magnetization, remanence and coercivity Nd_1.1_Fe_10_CoTiN_x_ alloy nitrogenated during 6 h and as a function of temperature. The specimen was not magnetically oriented.

**Table 1 molecules-26-03854-t001:** Lattice parameters, unit cell volume V, volume fraction and density ρ, for phases in the Nd_1.1_Fe_10_CoTiN_x_ alloys nitrogenated for 0, 3, 6, 9 and 12 h. All data were obtained through refinement of patterns.

Nitrogenation Time (h)	Phase	*a* (Å)	c (Å)	V (Å^3^)	Fraction (vol. %)	ρ (g/cm^3^)
**0**	1:12	8.589	4.808	354.7	95.7	7.579
	bcc	2.870			4.3	7.848
**3**	1:12 (a)	8.585	4.804	354.1	43.7	7.593
	1:12(b)	8.631	4.843	360.8	17.7	7.452
	1:12(c)	8.642	4.953	369.9	34.2	7.268
	bcc	2.870			4.4	7.843
**6**	1:12(a)	8.583	4.805	354.0	42.3	7.595
	1:12(b)	8.616	4.849	360.0	22.4	7.469
	1:12(c)	8.638	4.971	370.9	29.5	7.249
	bcc	2.871			5.8	7.840
**9**	1:12(a)	8.585	4.804	354.1	34.3	7.593
	1:12(b)	8.631	4.843	360.8	10.0	7.452
	1:12(c)	8.700	4.977	376.7	48.9	7.136
	bcc	2.873			6.8	7.820
**12**	1:12(a)	8.584	4.802	353.8	36.5	7.597
	1:12(b)	8.642	4.841	361.5	9.5	7.436
	1:12(c)	8.741	4.950	378.2	45.5	7.107
	bcc	2.871			8.5	7.840

Note: Uncertainties for the lattice parameters are ±0.001 Å and for unit cell volume is ±0.16 Å^3^.

**Table 2 molecules-26-03854-t002:** Magnetic properties of aligned and randomly oriented (in parenthesis) Nd_1.1_Fe_10_CoTiN_x_ powders nitrogenated for the indicated time.

Nitrogenation Time (h)	M_3T_ (Am^2^kg^−1^)	M_r_ (Am^2^kg^−1^)	μ_0_H_c_ (T)	(BH)_max_ (kJ·m^−3^)
**0**	122.83 (121.92)	108.21 (57.79)	0.294 (0.276)	81.07 (30.44)
**3**	128.80 (114.17)	116.34 (75.27)	0.504 (0.500)	113.74 (58.38)
**6**	128.97 (106.90)	116.14 (62.16)	0.518 (0.457)	111.41 (39.89)
**9**	126.38 (112.45)	113.24 (68.28)	0.395 (0.371)	96.40 (44.02)
**12**	126.97 (115.12)	112.87 (71.15)	0.316 (0.312)	87.85 (45.74)

Note: Uncertainties for M_3T_, M_r_, **μ**
_0_H_c_ and (BH)_max_ are ±0.20 Am^2^kg^−1^, ±0.14 Am^2^kg^−1^, ±0.001 T and ±0.35 kJ·m^−3^, respectively.

## Data Availability

The data that support the results of this research are available when the editorial office of this journal requires it.

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
