# Peer review of "Mechanochemical Synthesis and Nitrogenation of the Nd1.1Fe10CoTi Alloy for Permanent Magnet"

_molecules, 2021, doi:10.3390/molecules26133854_

Round 1

Reviewer 1 Report

This manuscript reports on structural and magnetic properties of the nanocrystalline Nd1.1Fe10CoTi compound. The work is interesting and it is worth to publish in Molecules. However the discussion of magnetic properties should be  supplemented with measurements of the temperature dependence of DC susceptibility ( in the temperature range of 10-300K). Otherwise, the discussion  on spin reorientation will be incomplete.

Author Response

Dear reviewer, 

Thank you for your comments and questions to clarify and improve the discussion of the manuscript. I attach a letter with the changes made to the manuscript and answers to the questions.

Sincerely,

Hugo Martínez Sánchez

Reviewer 2 Report

The authors used mechanochemical method to produce nanocrystalline Nd1.1Fe10CoTi  compound, is of a certain significance and value. The following modifications are recommended to be made:

1.explain how the density is determined and how the magnetic product (BH)max is calculated.

2.the permanent magnetic properties and the cost of this synthesis method should be compared with other method.  

Author Response

(The authors gave the same response as above.)
